# Posthoc privacy guarantees for collaborative inference with modified Propose-Test-Release

**Abhishek Singh**[1]**, Praneeth Vepakomma**[1]**, Vivek Sharma**[1,2,3,*]**, Ramesh Raskar**[1]

[1]MIT Media Lab, [2]MGH, Harvard Medical School, [3]Sony AI

## Abstract

Cloud-based machine learning inference is an emerging paradigm where users query by sending their data through a service provider who runs an ML model on that data and returns back the answer. Due to increased concerns over data privacy, recent works have proposed Collaborative Inference (CI) to learn a privacy-preserving encoding of sensitive user data before it is shared with an untrusted service provider. Existing works so far evaluate the privacy of these encodings through empirical reconstruction attacks. In this work, we develop a new framework that provides formal privacy guarantees for an arbitrarily trained neural network by linking its local Lipschitz constant with its local sensitivity. To guarantee privacy using local sensitivity, we extend the Propose-Test-Release (PTR) framework to make it tractable for neural network queries. We verify the efficacy of our framework experimentally on real-world datasets and elucidate the role of Adversarial Representation Learning (ARL) in improving the privacy-utility trade-off. The source code and other details are available at `tremblerz.github.io/posthoc`.

## 1 Introduction

While training ML models privately has seen tremendous progress[1, 45, 10, 26] in the last few years, protecting privacy during the inference phase remains elusive as these models get deployed by cloud-based service providers. Cryptographic techniques[43, 31, 40, 27] address this challenge by computing over encrypted data. To combat the high computational cost of encryption techniques, several recent works [66, 65, 5, 56, 57] have proposed Collaborative Inference (CI) – an alternate paradigm where users share a lower dimensional embedding of raw data where task-irrelevant information is suppressed. However, CI techniques are currently incompatible with formal privacy frameworks and their evaluation is currently based on empirical reconstruction attacks. The central issue in CI is the usage of Deep Neural Networks (DNNs) making it unsuitable for formal and worst-case privacy analysis. In this work, we take the first step towards a formal privacy framework for CI that will enable a more rigorous evaluation and a reliable understanding of the CI techniques.

The key aspect of any CI algorithm is an *obfuscator* (typically a DNN), which is trained to encode a user's private data such that an attacker can not reconstruct the original data from its encoding. Hence, CI techniques typically evaluate the privacy of their representations by empirically measuring the information leakage using a proxy adversary. However, existing works [57, 66, 18, 56] show that a proxy adversary's performance as a measure of protection could be unreliable because a future adversary can come up with a stronger attack. Alternately a few existing CI techniques have used theoretical tools [20, 69, 4, 68, 62, 5, 39] for measuring information leakage empirically. However, most of these works analyze specific obfuscation techniques and lack formal privacy definitions. Therefore, we first introduce a privacy definition applicable to CI by adopting a different threat model from that of differential privacy (DP)[12] because (as we explain in detail in Sec 2) protecting membership inference is at odds with achieving a non-trivial privacy-utility trade-off in CI.

---

*Part of this work was carried out while he was at MIT and MGH, Harvard Medical School.

37th Conference on Neural Information Processing Systems (NeurIPS 2023).

Our threat model for the reconstruction attack is motivated by the intuition that the obfuscator discloses coarse-grained and filters out fine-grained information for preventing reconstruction. We hypothesize that to do so, the *obfuscator* should act as a contractive mapping, and as a result, increases the *stability* of the (obfuscation) function in the local neighborhood of data. We formalize this intuition by introducing a privacy definition in the metric space of data and experimentally validate it in Sec 5. Our privacy definition is an instantiation of metric-DP by Chatzikokolakis et al. Instead of evaluating the global Lipschitz constant of DNNs, we evaluate the Lipschitz constant only in the local neighborhood of the user's sensitive data. We then extend the Propose-Test-Release (PTR)[11] framework to formalize our local neighborhood based measurement of the Lipschitz constant.

We adopt the PTR framework due to its posthoc and data-adaptive nature – as the privacy of encoding is evaluated in the neighborhood of the private user input and after the *obfuscator* has been trained. This approach is different from conventional privacy mechanisms where the mechanism is fixed and the perturbation does not depend on data. Existing seminal works in data-adaptive privacy such as Propose-Test-Release (PTR) [11], smooth sensitivity [41] and Lipschitz extensions [48] have exploited this posthoc approach to enhance the utility of the mechanisms. Data adaptive mechanisms are an important area of study as quoted in a recent workshop report on the challenges in Differential Privacy [9] - "*Going data-adaptive is a crucial step in bringing DP algorithms to an acceptable level of utility in applications.*". In the context of our work – the benefit of using the PTR framework is two-fold i) Utility - the *obfuscator* is expected to behave better at samples that match the data distribution from its training dataset are sampled, and ii) Computation - instead of computing global sensitivity, we only need to compute sensitivity in the small neighborhood around private data points. While the original PTR is intractable due to a well-known [9] expensive computation step, we design a new formulation that substitutes the expensive computation step with a tractable alternate.

Computational tractability is generally the main concern that makes CI incompatible with formal privacy because giving worst-case guarantees for *obfuscator* is generally difficult. While training and convergence guarantees for DNNs remain elusive, we find that the certified robustness community has made rapid progress in the past few years in giving worst-case guarantees during the inference stage of DNNs. Specifically, we utilize recent efficient formulations for the Lipschitz constant computation of DNNs [25, 55]. We measure the stability of the learned *obfuscator* model using the Lipschitz constant and link it with the local sensitivity under our privacy definition. We bridge the gap between the Lipschitz constant and our extended PTR framework by proposing a binary search algorithm that computes this Lipschitz constant multiple times in the neighborhood of sensitive data. This bridge makes our work amenable to any differentiable *obfuscator* with ReLU non-linearity.

The **scope** of our paper is to provide privacy guarantees against reconstruction attacks for existing CI techniques, *i.e.*, our goal is not to develop a new CI algorithm but rather to develop a formal privacy framework compatible with existing algorithms. A majority of the CI techniques protect either a sensitive attribute or reconstruction of the input. We only consider sensitive input in this work. Furthermore, we only focus on protecting the privacy of data during the inference stage, and assume that ML models can be trained privately. Our privacy definition and the mechanism are built upon $d_{\mathcal{X}}$-privacy [8] and PTR [11]. Existing instantiations of $d_{\mathcal{X}}$-privacy include geo-indistinguishability [2] and location-dependent privacy[32] that share a similar goal as ours of sharing coarse-grained information. Our work differs because we use DNN queries and high-dimensional datasets. We refer the reader to the supplementary for a detailed literature review.

In Sec 2 we begin with the preliminaries of DP and its variant for metric spaces. Then, we introduce the privacy definition relevant for CI in Sec 3. Next, we design our framework by extending PTR and proving its privacy guarantees in Sec 4. In Sec 5 we experimentally demonstrate the feasibility of our framework. We now give the **summary of our contributions:**

1. Proposing a privacy definition that formalizes reconstruction privacy in the context of CI. Protecting against membership inference (meaningfully) is not possible in CI and hence traditional DP can not be applied directly as discussed in Sec 2.

2. Extending the (PTR) algorithm to guarantee privacy using the local Lipschitz constant. The standard PTR can not be applied due to its intractability for queries beyond median and mode (see - Page 152, 2nd paragraph of Dwork et al. and Sec 4).

3. Experiments 1) to verify efficacy for both utility and computability, 2) ablation on different design choices, 3) empirical reconstruction attacks, and 4) identifying the role ARL techniques play in improving privacy-utility trade-off in CI.

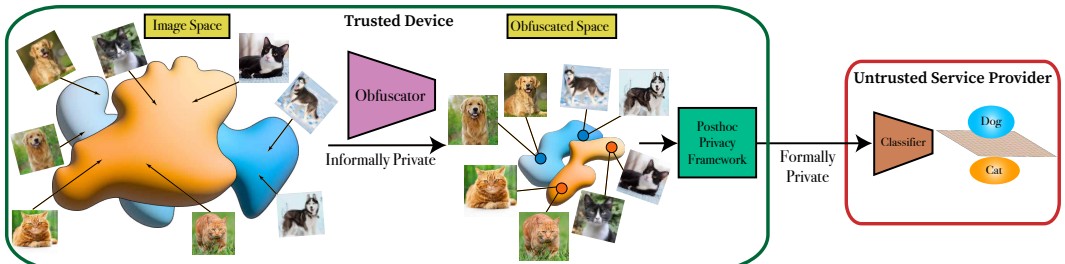

Figure 1: **Posthoc Privacy framework**: We project a high dimensional data instance to a lower dimensional embedding. The goal of the *embedder* is to measure a semantically relevant distance between different instances. The embedding is fed to the *Obfuscator* that compresses similar inputs in a small volume. In traditional ARL, the obfuscated instance is shared with the untrusted service provider without any formal privacy guarantee. In this work, by analyzing the stability of the obfuscator network, we perturb the obfuscated instance to provide a formal privacy guarantee.

## 2 Preliminaries

Differential privacy[12] is a widely used framework for answering a query, $f$, on a dataset $\mathbf{x} \in \mathcal{X}$ by applying a mechanism $\mathcal{M}(\cdot)$ such that the probability distribution of the output of the mechanism $\mathcal{M}(\mathbf{x})$ is *similar* regardless the presence or absence of any individual in the dataset $\mathbf{x}$. More formally, $\mathcal{M}$ satisfies $(\epsilon, \delta)$-DP if $\forall \mathbf{x}, \mathbf{x}' \in \mathcal{X}$ such that $d_H(\mathbf{x}, \mathbf{x}') \leq 1$, and for all (measurable) output $S$ over the range of $\mathcal{M}$

$$\mathbb{P}(\mathcal{M}(\mathbf{x}) \in S) \leq e^\epsilon \mathbb{P}(\mathcal{M}(\mathbf{x}') \in S) + \delta,$$

where $d_H$ is the hamming distance. This definition is based on a trusted central server model, where a trusted third party collects sensitive data and computes $\mathcal{M}(\mathbf{x})$ to share with untrusted parties. In *local*-DP[28], this model has been extended such that each user shares $\mathcal{M}(\mathbf{x})$, and the service provider is untrusted. Our threat model is the same as local DP. However, unlike local DP, data is not aggregated over multiple individuals. Specifically, every sample is evaluated independently by the service provider and there is no aggregation involved. For ex.– a user shares a face image to receive an age prediction from the service provider, here the answer to the query depends exactly on the user's input. With traditional local DP, it is "impossible" to achieve good utility and privacy at the same time because any two samples (no matter how different) are neighboring databases, more formally, $d_H(\mathbf{x}, \mathbf{x}') \leq 1, \ \forall \mathbf{x}, \mathbf{x}' \in \mathcal{X}$ for ML inference. Informally, this notion of neighboring databases under the standard DP definition would imply that the outcome for any two samples should be *almost* indistinguishable no matter how different the samples are. This privacy definition could be too restrictive for our ML inference application where the data instance necessarily needs a certain degree of distinguishability to obtain utility from the service provider. This observation is formalized in the impossibility result of instance encoding[7] for private learning. To subside this fundamental conflict between the privacy definition and our application, we look at the definition of $d_\mathcal{X}$-privacy[8] that generalizes the DP definition to a general distance metric beyond hamming distance as follows:

$$\mathbb{P}(\mathcal{M}(\mathbf{x}) \in S) \leq e^{d_\mathcal{X}(\mathbf{x}, \mathbf{x}')} \mathbb{P}(\mathcal{M}(\mathbf{x}') \in S), \tag{1}$$

here $d_\mathcal{X}(\mathbf{x}, \mathbf{x}')$ is a function that gives a level of indistinguishability between two datasets $\mathbf{x}$ and $\mathbf{x}'$. DP can be viewed as a special case of $d_\mathcal{X}$-privacy by keeping $d_\mathcal{X}(\mathbf{x}, \mathbf{x}') = \epsilon d_H(\mathbf{x}, \mathbf{x}')$. Choosing a different distance metric yields a stronger or weaker privacy guarantee. Due to the impossibility of achieving both privacy and utility with hamming distance, we move towards a weaker privacy definition by only focusing on achieving privacy against reconstruction attacks instead of membership inference. Intuitively, we choose a distance metric that provides indistinguishability only within a neighborhood of similar *looking* samples.

## 3 Privacy Definition and Threat Model

To formalize reconstruction privacy, we hypothesize that semantically similar points are close to each other on a data manifold, i.e. semantically similar samples are closer in a space where distance is defined as geodesic on the data manifold. Therefore, one way to bound the reconstruction of $\mathbf{x}$

is by making it indistinguishable among semantically similar points. The extent of reconstruction privacy would therefore depend upon the radius of the neighborhood. We formalize it with a privacy parameter $R$ that allows a user to control how big this indistinguishable neighborhood should be.

**Definition 1.** *A mechanism* $\mathcal{M}$ *satisfies* $(\epsilon, \delta, R)$*-reconstruction privacy if* $\forall \mathbf{x}, \mathbf{x}' \in \mathcal{X}$ *s.t.* $d_{\mathcal{X}}(\mathbf{x}, \mathbf{x}') \leq R$ *and* $S \subseteq Range(\mathcal{M})$

$$\mathbb{P}(\mathcal{M}(\mathbf{x}) \in S) \leq e^{\epsilon}\mathbb{P}(\mathcal{M}(\mathbf{x}') \in S) + \delta. \tag{2}$$

Note that the above equation is exactly the same as $(\epsilon, \delta)$-DP except for the definition of neighboring databases. In this way, our privacy definition can be seen as a mix of standard DP and $d_{\mathcal{X}}$-privacy.

## 3.1 Threat and Attack model

Our threat and attack model is the same as previous collaborative inference works [66, 65]. Specifically, we focus on attackers that aim to reconstruct the original sample from its encoding. That is, given an encoding $\mathcal{M}(\mathbf{x})$ obtained from a sample $\mathbf{x}$, an attacker will attempt to recover the original sample ($\mathbf{x}$). Our threat model is the same as local DP. Specifically, all side information including access to the obfuscator model, embedder model, and the framework in Fig 1 is known to the adversary and it can interact with them as much as it wants. The attacker can use this information to improve its attack algorithm. We (informally) note that an attacker always has a prior about raw data $\mathbb{P}(\mathbf{x})$ and it will update its belief based on the posterior $\mathbb{P}(\mathbf{x}|\mathcal{M}(\mathbf{x}))$. The goal of our privacy mechanism is to ensure that the distance between the prior and posterior distribution is bounded. Our definition indeed enjoys such an interpretation due to being an instantiation of $d_{\mathcal{X}}$-privacy. A more formal description of this Bayesian formulation is given by Chatzikokolakis et al. in Theorem 4. We emulate such attackers in Sec 5 by training an ML model on a dataset of $\mathcal{M}(\mathbf{x})$ as input and $\mathbf{x}$ as output, then we evaluate its reconstruction error on encodings of previously unseen $\mathcal{M}(\mathbf{x})$.

## 3.2 Comparison with Differential Privacy

Conceptually, the usage of hamming distance in DP for neighboring databases provides a level of protection at the sample level. Such a privacy requirement can be at odds with the goal of prediction that necessarily requires discrimination between samples belonging to different concept classes. We relax this dichotomy by changing the distance metric to account for reconstruction privacy. Intuitively, an attacker can only obtain an accurate reconstruction of data up to a neighborhood in the space of data. Therefore, the size of the neighborhood is a privacy parameter $R$ such that a higher value of $R$ provides higher privacy. This privacy parameter is equivalent to the *group size* [12] used often in the DP literature. By default, this value is kept at 1 in DP but can be kept higher if correlated individuals (multiple samples, family) are present in a database instead of a single individual. We state the equivalence between the group privacy definition and standard DP informally -
**Lemma 2.2** in [60]: Any $(\epsilon, \delta)$-differentially private mechanism is $(R\epsilon, Re^{(R-1)\epsilon}\delta)$-differentially private for groups of size $R$.
This lemma also applies to our proposed definition. However, we emphasize that privacy parameters of $(\epsilon, \delta)$-DP mechanism can not be compared trivially with a $(\epsilon, \delta, R)$-reconstruction privacy mechanism because same value of $\epsilon$ and $\delta$ provide different levels of protection due to different definitions of neighboring databases. We experimentally demonstrate this claim in Sec. 5.

## 3.3 Choice of $d_{\mathcal{X}}(\mathbf{x}, \mathbf{x}')$ and its impact on the privacy guarantees offered

Our privacy definition is an instantiation of $d_{\mathcal{X}}(\mathbf{x}, \mathbf{x}')$ that permits a broad class of distance metrics for the privacy definition. Since the goal of our work is to protect against reconstruction - $\ell_1$ or $\ell_2$ distance is a reasonable choice. However, the space over which $\ell$ norm should be considered is a challenging question. Existing papers have used metrics such as $\ell_1$ or $\ell_2$ norm in the raw data space [21, 6, 17, 3] for guaranteeing reconstruction privacy when training ML models. However, for high-dimensional datasets typically considered in CI such as images, guaranteeing reconstruction in ambient dimensions can lead to unintended privacy guarantee. In our work, we consider pre-processing the samples by projecting them into an embedding space, where samples that are semantically closer and farther have $\ell$ norm smaller and higher respectively. We evaluate the privacy-utility trade-off in our experiments for both ambient space and embedding space.

## 4 Privacy Mechanism

Our goal is to design a framework that can provide a formal privacy guarantee for encodings informally privatized in CI. However, CI algorithms use non-linear neural networks trained on non-convex objectives making it difficult to perform any worst-case analysis. Therefore, we take a posthoc approach where we reason about privacy after the model is trained. Specifically, we apply the PTR mechanism by [11]. Applying PTR directly to our query (ARL) is not computationally feasible because PTR requires estimating local sensitivity at multiple points whereas evaluating the local sensitivity of a neural network query is not even feasible at a single point. Therefore, we design a tractable variant of PTR that utilizes the local Lipschitz constant estimator to compute privacy related parameters. We refer the reader to supplementary for background on the Lipschitz constant estimation and PTR.

CI algorithms consist of three computational blocks in the training stage: 1) *obfuscator* ($f(\cdot)$) that generates an (informally private) representation ($\tilde{\mathbf{z}}$) of data, 2) *proxy adversary* that reconstructs the data from the representation produced by the *obfuscator*, and 3) *classifier* that performs the given task using the obfuscated representation. The *classifier* and *proxy adversary* are trained to minimize the task loss and reconstruction loss, respectively. The *obfuscator* is trained to minimize the task loss but maximize the reconstruction loss. This setup results in a min-max optimization where the trade-off between task performance and reconstruction is controlled by a hyper-parameter $\alpha$. Note that a few CI techniques[42, 44, 61] do not require a proxy adversary but still learn an obfuscator model using other regularizers.

At a high level, our framework applies the mechanism $\mathcal{M}$ such that the final released data $\hat{\mathbf{z}} = \mathcal{M}(\mathbf{x})$ has a privacy guarantee discussed in Eq 2. Like PTR, we start with a proposal ($\Delta_{LS}^p$) on the upper bound of the local sensitivity of $\mathbf{x}$. To test the correctness of $\Delta_{LS}^p$, we compute the size of the biggest possible neighborhood such that the local Lipschitz constant of the *obfuscator* network in that neighborhood is smaller than the proposed bound. Next, we privately verify the correctness of the proposed bound. We do not release the data (denoted by $\perp$) if the proposed bound is invalid. Otherwise, we perturb the data using the Laplace mechanism. Next, we discuss the framework and privacy guarantees in more detail.

**Global Sensitivity and Lipschitz constant** of a query $f : \mathcal{X} \to \mathcal{Y}$ are related in the $d_{\mathcal{X}}$-privacy framework. Global sensitivity of a query $f(\cdot)$ is the smallest value of $\Delta$ (if it exists) such that $\forall \mathbf{x}, \mathbf{x}' \in \mathcal{X}, d_{\mathcal{Y}}(f(\mathbf{x}), f(\mathbf{x}')) \leq \Delta d_{\mathcal{X}}(\mathbf{x}, \mathbf{x}')$. While global sensitivity is a measure over all possible pairs of data in the data domain $\mathcal{X}$, local sensitivity ($\Delta_{LS}$) is defined with respect to a given dataset $\mathbf{x}$ such that $\forall \mathbf{x}' \in \mathcal{X}, d_{\mathcal{Y}}(f(\mathbf{x}), f(\mathbf{x}')) \leq \Delta_{LS}(\mathbf{x}) d_{\mathcal{X}}(\mathbf{x}, \mathbf{x}')$. We integrate the notion of similarity in a neighborhood (motivated in Sec 2) by defining the local sensitivity of a neighborhood $\mathcal{N}(\mathbf{x}, R)$ around $\mathbf{x}$ of radius $R$ where $\mathcal{N}(\mathbf{x}, R) = \{\mathbf{x}' | d_{\mathcal{X}}(\mathbf{x}, \mathbf{x}') \leq R, \forall \mathbf{x}' \in \mathcal{X}\}$. Therefore, the local sensitivity of query $f$ on $\mathbf{x}$ in the $R$-neighborhood is defined $\forall \mathbf{x}' \in \mathcal{N}(\mathbf{x}, R)$ such that

$$d_{\mathcal{Y}}(f(\mathbf{x}), f(\mathbf{x}')) \leq \Delta_{LS}(\mathbf{x}, R) d_{\mathcal{X}}(\mathbf{x}, \mathbf{x}'). \tag{3}$$

We note that if $d_{\mathcal{X}}$ is hamming distance and $R$ is 1 then this formulation is exactly same as local sensitivity in $\epsilon$-DP[12]. The equation above can be re-written as:

$$\Delta_{LS}(\mathbf{x}, R) = \sup_{\mathbf{x}' \in \mathcal{N}(\mathbf{x}, R)} \frac{d_{\mathcal{Y}}(f(\mathbf{x}), f(\mathbf{x}'))}{d_{\mathcal{X}}(\mathbf{x}, \mathbf{x}')}. \tag{4}$$

This formulation of local sensitivity is similar to the definition of the local Lipschitz constant. The local Lipschitz constant $\mathcal{L}$ of $f$ for a given open neighborhood $\mathcal{N} \subseteq \mathcal{X}$ is defined as follows:

$$\mathcal{L}^{\alpha, \beta}(f, \mathcal{N}) = \sup_{\mathbf{x}', \mathbf{x}'' \in \mathcal{N}} \frac{||f(\mathbf{x}') - f(\mathbf{x}'')||_{\alpha}}{||\mathbf{x}' - \mathbf{x}''||_{\beta}} \quad (\mathbf{x}' \neq \mathbf{x}'') \tag{5}$$

We note that while the local sensitivity of $\mathbf{x}$ is described around the neighborhood of $\mathbf{x}$, the Lipschitz constant is defined for every possible pair of points in a given neighborhood. Therefore, in Lemma 4.1 we show that the local Lipschitz in the neighborhood of $\mathbf{x}$ is an upper bound on the local sensitivity.

**Lemma 4.1.** *For a given $f$ and for $d_{\mathcal{Y}} \leftarrow \ell_{\alpha}$ and $d_{\mathcal{X}} \leftarrow \ell_{\beta}$, $\Delta_{LS}(\mathbf{x}) \leq \mathcal{L}(f, \mathcal{N}(\mathbf{x}, R))$. Proof in Supplementary.*

Since local sensitivity is upper bounded by the Lipschitz constant, evaluating the Lipschitz constant suffices as an alternative to evaluating local sensitivity.

**Lower bound on testing the validity of $\Delta_{LS}^p$:** The PTR algorithm [11] suggests a proposal on the upper bound ($\Delta_{LS}^p$) of local sensitivity and then finds the distance between the given dataset ($\mathbf{x}$) and the closest dataset for which the proposed upper bound is not valid. Let $\gamma(\cdot)$ be a distance query and $\Delta_{LS}(\mathbf{x})$ be the local sensitivity defined as per the DP framework with respect to $\mathbf{x}$ such that

$$\gamma(\mathbf{x}) = \min_{\mathbf{x}' \in \mathcal{X}} \{d_H(\mathbf{x}, \mathbf{x}') \ \ s.t. \ \ \Delta_{LS}(\mathbf{x}') > \Delta_{LS}^p\}. \tag{6}$$

For our privacy definition, the query $\gamma(\mathbf{x}, R)$ can be formulated as follows:

$$\gamma(\mathbf{x}, R) = \min_{\mathbf{x}' \in \mathcal{X}} \{d_{\mathcal{X}}(\mathbf{x}, \mathbf{x}') \ \ s.t. \ \ \Delta_{LS}(\mathbf{x}', R) > \Delta_{LS}^p\}. \tag{7}$$

We note that keeping $d_{\mathcal{X}} = d_H$ and $R = 1$, makes the $\gamma$ query exactly same as defined in the eq 6. Generally, computing $\gamma(\cdot)$ is intractable due to local sensitivity estimation required for every $\mathbf{x}' \in \mathcal{X}$ (which depends upon a non-linear neural network in our case). We emphasize that this step is intractable at two levels, first, we require estimating the local sensitivity of a neural network query. Second, we require this local sensitivity over all samples in the data domain. Therefore, we make it tractable by computing a lower bound over $\gamma(\mathbf{x}, R)$ by designing a function $\phi(\cdot)$ s.t. $\phi(\mathbf{x}, R) \leq \gamma(\mathbf{x}, R)$. Intuitively, $\phi(\cdot)$ finds the largest possible neighborhood around $\mathbf{x}$ such that the local Lipschitz constant of the neighborhood is smaller than the proposed local sensitivity. Because the subset of points around $\mathbf{x}$ whose neighborhood does not violate $\Delta_{LS}^p$ is half of the size of the original neighborhood in the worst case, we return half of the size of neighborhood as the output. We describe the algorithm procedurally in the supplementary material. More formally,

$$\phi(\mathbf{x}, R) = \frac{1}{2} \cdot \arg\max_{R' \geq R} \{\mathcal{L}(f, \mathcal{N}(\mathbf{x}, R')) \leq \Delta_{LS}^p\}$$

If there is no solution to the equation above, then we return 0.

**Lemma 4.2.** $\phi(\mathbf{x}, R) \leq \gamma(\mathbf{x}, R)$. *Proof in Supplementary.*

**Privately testing the lower bound:** The next step in the PTR algorithm requires testing if $\gamma(\mathbf{x}) \leq \ln(\frac{1}{\delta}/\epsilon)$. If the condition is true, then no-answer ($\perp$) is released instead of data. Since the $\gamma$ query depends upon $\mathbf{x}$, PTR privatizes it by applying laplace mechanism, i.e. $\hat{\gamma}(\mathbf{x}) = \gamma(\mathbf{x}) + \mathsf{Lap}(1/\epsilon)$. The query has a sensitivity of 1 since the $\gamma$ could differ at most by 1 for any two neighboring databases. In our framework, we compute $\phi(\mathbf{x}, R)$ to lower bound the value of $\gamma(\mathbf{x}, R)$. Therefore, we need to privatize the $\phi$ query. We prove that for distance metrics in $d_{\mathcal{X}}$-privacy, the global sensitivity of the $\phi(\mathbf{x})$ query is 1.

**Lemma 4.3.** *The query $\phi(\cdot)$ has a global sensitivity of 1, i.e. $\forall \mathbf{x}, \mathbf{x}' \in \mathcal{X}, d_{abs}(\phi(\mathbf{x}, R), \phi(\mathbf{x}', R)) \leq d_{\mathcal{X}}(\mathbf{x}, \mathbf{x}')$. Proof in Supplementary.*

After computing $\phi(\mathbf{x}, R)$, we add noise sampled from a laplace distribution, i.e. $\hat{\phi}(\mathbf{x}, R) = \phi(\mathbf{x}, R) + \mathsf{Lap}(R/\epsilon)$. Next, we check if $\hat{\phi}(\mathbf{x}, R) \leq \ln(\frac{1}{\delta}) \cdot R/\epsilon$, then we release $\perp$, otherwise we release $\hat{\mathbf{z}} = f(g(\mathbf{x})) + \mathsf{Lap}(\Delta_{LS}^p/\epsilon)$. Next, we prove that the mechanism $\mathcal{M}_1$ described above satisfies *reconstruction-privacy*.

**Theorem 4.4.** *Mechanism $\mathcal{M}_1$ satisfies uniform $(2\epsilon, \delta/2, R)$-reconstruction privacy Eq. 2, i.e. $\forall \mathbf{x}, \mathbf{x}' \in \mathcal{X}, s.t. \ d_{\mathcal{X}}(\mathbf{x}, \mathbf{x}') \leq R$*

$$\mathbb{P}(\mathcal{M}(\mathbf{x}) \in S) \leq e^{2\epsilon} \mathbb{P}(\mathcal{M}(\mathbf{x}') \in S) + \frac{\delta}{2} \tag{8}$$

Proof Sketch: Our proof is similar to the proof for the PTR framework[12] except the peculiarity introduced due to our metric space formulation. First, we show that not releasing the answer ($\perp$) satisfies the privacy definition. Next, we divide the proof into two parts, when the proposed bound is incorrect (i.e. $\Delta_{LS}(\mathbf{x}, R) > \Delta_{LS}^p$) and when it is correct. Let $\hat{R}$ be the output of query $\phi$.

$$\frac{\mathbb{P}[\hat{\phi}(\mathbf{x}, R) = \hat{R}]}{\mathbb{P}[\hat{\phi}(\mathbf{x}', R) = \hat{R}]} = \frac{\exp(-(\frac{|\phi(\mathbf{x}, R) - \hat{R}|}{R} \cdot \epsilon))}{\exp(-(\frac{|\phi(\mathbf{x}', R) - \hat{R}|}{R} \cdot \epsilon))}$$

$$\leq \exp(|\phi(\mathbf{x}', R) - \phi(\mathbf{x}, R)| \cdot \frac{\epsilon}{R}) \leq \exp(d_{\mathcal{X}}(\mathbf{x}, \mathbf{x}') \cdot \frac{\epsilon}{R}) \leq \exp(\epsilon)$$

| | MNIST ($\epsilon = 0$, 0.10), ($\epsilon = \infty$, 0.93) | | | | | FMNIST ($\epsilon = 0$, 0.10), ($\epsilon = \infty$, 0.781) | | | | | UTKFace ($\epsilon = 0$, 0.502), ($\epsilon = \infty$, 0.732) | | | | |
|---|---|---|---|---|---|---|---|---|---|---|---|---|---|---|---|
| | Informal | $\epsilon=1$ | $\epsilon=2$ | $\epsilon=5$ | $\epsilon=10$ | Informal | $\epsilon=1$ | $\epsilon=2$ | $\epsilon=5$ | $\epsilon=10$ | Informal | $\epsilon=1$ | $\epsilon=2$ | $\epsilon=5$ | $\epsilon=10$ |
| Encoder | **0.93** | 0.428 | 0.673 | 0.883 | **0.921** | 0.779 | 0.228 | 0.355 | 0.605 | **0.722** | 0.724 | 0.617 | 0.673 | 0.717 | 0.721 |
| ARL | 0.917 | 0.329 | 0.532 | 0.792 | 0.882 | 0.747 | 0.214 | 0.319 | 0.557 | 0.685 | 0.71 | 0.605 | 0.649 | 0.691 | 0.707 |
| C | 0.926 | 0.443 | 0.684 | 0.881 | 0.917 | **0.781** | 0.158 | 0.225 | 0.422 | 0.608 | **0.73** | 0.623 | 0.673 | **0.718** | **0.724** |
| N | 0.923 | 0.279 | 0.496 | 0.816 | 0.902 | 0.559 | 0.136 | 0.177 | 0.310 | 0.462 | 0.725 | 0.614 | 0.667 | 0.708 | 0.715 |
| ARL-C | 0.896 | 0.424 | 0.648 | 0.839 | 0.883 | 0.761 | 0.196 | 0.314 | 0.537 | 0.682 | 0.709 | 0.632 | 0.684 | 0.70 | 0.705 |
| ARL-N | 0.88 | 0.118 | 0.139 | 0.21 | 0.325 | 0.717 | 0.294 | 0.467 | 0.657 | 0.705 | 0.71 | 0.628 | 0.674 | 0.701 | 0.708 |
| C-N | 0.929 | 0.353 | 0.574 | 0.844 | 0.913 | 0.774 | 0.161 | 0.224 | 0.411 | 0.599 | 0.727 | 0.616 | 0.671 | 0.712 | 0.722 |
| ARL-C-N | 0.921 | **0.514** | **0.751** | **0.891** | 0.912 | 0.706 | **0.371** | **0.554** | **0.678** | 0.695 | 0.712 | **0.650** | **0.690** | 0.700 | 0.700 |

Table 1: **Performance comparison for different baselines:** Our posthoc framework enables comparison between different obfuscation techniques by fixing the privacy budget ($\epsilon$). First four rows are different approaches to protect against data reconstruction and the remaining rows below are combinations of different approaches. The top row refers to the accuracy corresponding to different datasets under two extremes of epsilons. ARL refers to widely used adversarial representation learning approach for regularizing representation based on a proxy attacker[35, 38, 65, 56]. Contrastive refers to contrastive learning based informally privatizing mechanism introduced in [44].

Therefore, using the post-processing property - $\mathbb{P}[\mathcal{M}(\mathbf{x}) = \bot] \leq e^\epsilon \mathbb{P}[\mathcal{M}(\mathbf{x}') = \bot]$. Here, the first inequality is due to triangle inequality, the second one is due to Lemma 4.3 and the third inequality follows from $d_\mathcal{X}(\mathbf{x}, \mathbf{x}') \leq R$. Note that when $\Delta_{LS}(\mathbf{x}, R) > \Delta_{LS}^p$, $\phi(\mathbf{x}, R) = 0$. Therefore, the probability for the test to release the answer in this case is

$$\mathbb{P}[\mathcal{M}(\mathbf{x}) \neq \bot] = \mathbb{P}[\phi(\mathbf{x}, R) + \mathsf{Lap}(\frac{R}{\epsilon}) > \log(\frac{1}{\delta}) \cdot \frac{R}{\epsilon}] = \mathbb{P}[\mathsf{Lap}(\frac{R}{\epsilon}) > \log(\frac{1}{\delta}) \cdot \frac{R}{\epsilon}]$$

Based on the CDF of Laplace distribution, $\mathbb{P}[\mathcal{M}(\mathbf{x}) \neq \bot] = \frac{\delta}{2}$. Therefore, if $\Delta_{LS}(\mathbf{x}, R) > \Delta_{LS}^p$, for any $S \subseteq \mathbb{R}^d \cup \bot$ in the output space of $\mathcal{M}$

$$\mathbb{P}[\mathcal{M}(\mathbf{x}) \in S] = \mathbb{P}[\mathcal{M}(\mathbf{x}) \in S \cap \{\bot\}] + \mathbb{P}[\mathcal{M}(\mathbf{x}) \in S \cap \{\mathbb{R}^d\}]$$

$$\leq e^\epsilon \mathbb{P}[\mathcal{M}(\mathbf{x}') \in S \cap \{\bot\}] + \mathbb{P}[\mathcal{M}(\mathbf{x}) \neq \bot] \leq e^\epsilon \mathbb{P}[\mathcal{M}(\mathbf{x}') \in S] + \frac{\delta}{2}$$

If $\Delta_{LS}(\mathbf{x}, R) \leq \Delta_{LS}^p$ then the mechanism is a composition of two $(\epsilon, \delta, R)$-reconstruction private algorithm where the first algorithm ($\phi(\mathbf{x}, R)$) is $(\epsilon, \delta/2, R)$-reconstruction private and the second algorithm is $(\epsilon, 0, R)$-reconstruction private. Using composition, the algorithm is $(2\epsilon, \delta/2, R)$-reconstruction private. We describe $\mathcal{M}_1$ step by step in the algorithm in supplementary. To summarize, we designed the posthoc privacy framework that extends the PTR framework by making it tractable to get $(\epsilon, \delta, R)$-reconstruction privacy. The exact local Lipschitz constant of the neural network based obfuscator is estimated using mixed-integer programming based optimization developed by [25].

**Computational feasibility**: Our key idea is to add extra computation on the client side to turn informally private representations to their formal counterpart. This extra computational cost is due to the estimation of the local Lipschitz constant of the *obfuscator* network. However, the following two factors of our framework make it practically feasible -

1. We compute the local Lipschitz constant (i.e. in a small neighborhood around a given point): Our extension of the propose-test-release framework only requires us to operate in a small local neighborhood instead of estimating the global Lipschitz constant which would be much more computationally expensive.

2. Low number of parameters for obfuscator: Instead of estimating the Lipschitz constant of the whole prediction model, we only require estimation of the obfuscator - a neural network that has a significantly lower number of parameters in comparison to the prediction model.

We experimentally validate both of the above benefits in Sec 5. The fact that the local Lipschitz constant is being computed over the same obfuscator allows room for optimizing performance by caching. Our goal is to demonstrate the feasibility of bridging formal privacy guarantees and CI mechanisms, hence, we did not explore such performance speedups.

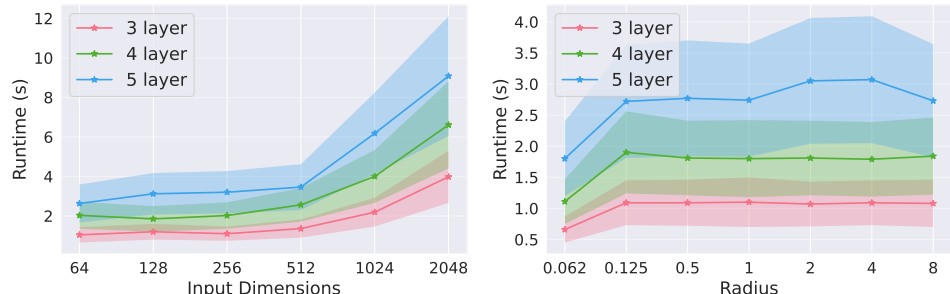

Figure 2: **Runtime evaluation of local lipschitz computation** for different (a) neighborhood radius, (b) input dimensions, and (c) number of layers. While the runtime increases exponentially with dimensions, it plateaus with increase in neighborhood radius. Since the input dimensions are same as embedding dimensions it makes the algorithm favorable to our analysis.

## 5 Experiments

**Experimental Setup:** We evaluate different aspects of our proposed framework - i) **E1**: comparison between different adversarial appraches, ii) **E2**: comparison with local differential privacy (LDP), iii) **E3**: computational tractability of our proposed framework, and iv) **E4**: investigating role of ARL in achieving privacy. We use MNIST[34], FMNIST[64] and UTKFace[67] dataset for all experiments. All of them contain samples with high ambient dimensions (MNIST-784, FMNIST-784 and UTKFace-4096). We use a deep CNN based $\beta$-VAE[22] for the *pre-processing*. We use LipMip[25] for computing Lipschitz constant over $\ell_\infty$ norm in the input space and $\ell_1$ norm in the output space. Our baselines include a simple *Encoder* that projects the data to smaller dimensions using a neural network. This encoder type approach has been used in the literature as Split Learning [19]. We include widely used ARL techniques [65, 56, 38, 35, 66] and adversarial contrastive learning[44] which we denote as C. We use noisy regularization (denoted by N) to improve classifier performance. We refer the reader to supplementary for a detailed experimental setup, codebase, and hyper-parameters.

**E1: Privacy-Utility Trade-off:** Since our framework enables comparison between different obfuscation techniques under same privacy budget, we evaluate test set accuracy as utility in Table 1. Our results indicate that ARL complemented with contrastive and noise regularization helps in attaining overall best performance among all possible combinations.

**E2: Comparison between ARL and LDP:** While $\epsilon$-LDP definition provides a different privacy guarantee than our proposed privacy definition, for reference we compare the performance between LDP and CI and report results in supplementary. We note that for low values of $\epsilon$, LDP techniques do not yield any practical utility. This corroborates with the impossibility result of instance encoding [7] and our discussion in Sec 2 about the inapplicability of traditional DP in the context of CI.

**E3: Computational feasibility:** We report end-to-end runtime evaluation on a CPU-based client and achieve **2 sec/image** (MNIST) and **3.5 sec/image** (UTKFace). While plenty of room exists for optimizing this runtime, we believe current numbers serve as a reasonable starting point for providing formal privacy in ARL. As discussed in Sec 4, we compare the computation time of the *obfuscator* across three factors relevant to our setup - i) Dimensionality of the input, ii) Size of the neighborhood, and iii) Number of layers in the *Obfuscator*. We report the results in Fig. 2.

**E4: What role does ARL play in achieving privacy?** We investigate the contribution of adversarial training in improving the privacy-utility trade-off. We train *obfuscator* models with different values of $\alpha$ (weighing factor) for adversarial training. Our results in Fig 3 indicate that higher weighing of adversarial regularization reduces the local Lipschitz constant, hence reducing the local sensitivity of the neural network. Furthermore, for high values of $\alpha$, the change in the local Lipschitz constant reduces significantly for different sizes ($R$) of the neighborhood. These two observations can potentially explain that ARL improves reconstruction privacy by reducing the sensitivity of the *obfuscator*. However, as we observe in Table 1, the classifier can reduce its utility if ARL is not complemented with noisy and contrastive regularization. We believe this finding could be of independent interest to the adversarial defense community.

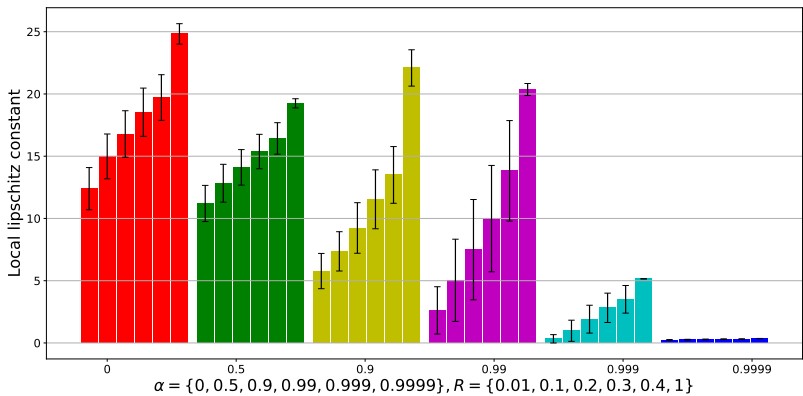

Figure 3: **Local sensitivity comparison for different values of** $\alpha$: The five bars for each $\alpha$ represent different neighborhood radii. Increase in the value of $\alpha$ decreases the local Lipschitz constant (upper bound on local sensitivity) indicating lesser amount of noise to be added for the same level of privacy.

## 6  Discussion

**How to select privacy parameter $R$?** One of the key difference between $(\epsilon, \delta, R)$-neighborhood privacy and $(\epsilon, \delta)$-DP is the additional parameter $R$. The choice of $R$ depends upon the neighborhood in which a user wishes to get an $\epsilon$ level of indistinguishability. We discussed the equivalence of $R$ and $k$ in group Differential Privacy in Sec 3.2. To understand the role of $R$, we perform reconstruction attacks on privatized encoding obtained from our framework by training an ML model to reconstruct original images. We compare reconstruction results for different values of $\epsilon$ and $R$ on four distinct metrics for images and report in Supplementary. To assess the level of indistinguishability, we project the original images into embedding space and sample points from the boundary of neighborhoods of different $R$. We observe that as the boundary of the neighborhood increases, the images become perceptually different from the original image. For extremely large radii, the images change significantly enough that their corresponding label may change too. Such visualizations can be used to semantically understand different values of $R$.

**How to propose $\Delta_{LS}^p$?** Our framework requires a proposal on the upper bound of local sensitivity in a private manner. One possible way to obtain $\Delta_{LS}^p$ is by using the Lipschitz constant of training data samples used in training the *obfuscator*. We choose $\Delta_{LS}^p$ by computing the mean ($\mu$) and standard deviation ($\sigma$) of local sensitivity on the training dataset (assumed to be known under our threat model), then we keep $\Delta_{LS}^p = \mu + n * \sigma$ where $n$ allows a trade-off between the likelihood of releasing the samples under PTR and adding extra noise to data. We used $n = 3$ in our experiments. Since empirically, the value of local sensitivity appears to be following a gaussian, using confidence interval serves as a good proxy. Fig 3 shows that for higher values of $\alpha$, the variability in the local Lipschitz constant decreases indicating the validity of the bound would hold for a large number of samples.

**Limitations:** i) Since we utilize the PTR framework, outlier samples may not get released due to high sensitivity, this is expected since these outlier samples are likely to be misclassified anyway. ii) Lipschitz constant computation is limited to ReLU based DNNs, therefore more sophisticated obfuscator architectures are currently not compatible with our proposed framework and we leave it as part of the future work. iii) Choosing the privacy parameter ($R$) could be challenging for certain datasets and might vary based on user preferences. We believe the choice of such a parameter would depend upon the context in which CI is being applied.

## 7  Related Work

**Collaborative Inference** techniques aim to *learn* a task-oriented privacy preserving encoding of data. Majority of the works in this area either protect against sensitive attribute leakage [20, 50, 5, 36] or input reconstruction [52, 56, 39, 35, 38]. These techniques usually evaluate their privacy using empirical attacks since the mechanism is learned using gradient based min-max optimization making it infeasible for the worst-case privacy analysis. The problem is exacerbated in the context of input reconstruction because of the lack of formal definition for reconstruction privacy. The goal of our work is to create a framework that make the existing techniques amenable to formal privacy analysis.

While theoretical analyses [68, 69, 51] of ARL objectives have identified fundamental trade-offs between utility and attribute leakage, they are difficult to formalize as a worst-case privacy guarantee especially for deep neural networks.

**Privacy definitions** that extend the DP definition to incorporate some of its restrictions [29] include $d_{\mathcal{X}}$-Privacy [8], and Pufferfish [30]. Our privacy definition is a specific instantiation of the $d_{\mathcal{X}}$-Privacy [8] framework that extends DP to general metric spaces. Our instantiation is focused on reconstruction privacy for individual samples instead of membership inference attacks [13]. Existing works in DP for reconstruction attacks [6, 58] focus on protecting training data from an attacker that has access to model weights (whitebox attacker) or output of model queries (blackbox attacker). In contrast, our work only focuses on the privacy of the data used during the prediction stage and not training data.

**Lipschitz constant** estimation for neural networks has been used to guarantee network's stability to perturbations. Existing works either provide an upper bound [63, 33, 14], exact Lipschitz constant [25, 24] or Lipschitz constant regularization [53, 23] during the training stage. Some existing works have explored the relationship between adversarial robustness and DP model training [46, 47, 59]. We utilize similar ideas of perturbation stability but for privacy. Shavit and Gjura [54] use Lipschitz neural networks [16] to learn a private mechanism design for summary statistics such as median, however their mechanism design lack privacy guarantee.

**Posthoc approach to privacy** applies privacy preserving mechanism in a data dependent manner. Smooth sensitivity [41] and PTR [11] reduce the noise magnitude because the local sensitivity is only equal to global sensitivity in the worst case and not average case. Privacy odometer [49], Ex-post privacy loss [37] and Rényi privacy filter [15] track privacy loss as the query is applied over data. Our works builds upon the PTR framework in order to give high privacy for less sensitive data. However, as we show in Sec 4, our framework reformulates the PTR algorithm to make it tractable under our setup.

# 8 Conclusion

In this work we take a first step towards bridging empirical techniques in ARL and formal privacy mechanisms. The posthoc nature of our framework allows reasoning about privacy without any modification to the obfuscation algorithm, hence, making it easy to integrate for benchmarking existing and future techniques. We introduced a privacy definition that formalizes ARL and designed a corresponding privacy mechanism. An exciting future direction for extending our framework is to integrate NNs for designing more effective privacy mechanisms for other sophisticated queries beyond CI. Data adaptive mechanisms are an exciting area in DP and our improvement to PTR can be potentially applied for other queries where PTR might be intractable currently.

# 9 Acknowledgements and Disclosure of funding

We thank Mohammad Mohammadi Amiri for his valuable feedback on the early draft. This work is supported by NSF award number 1729931.

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
