# Supplementary Material: Posthoc privacy guarantees for collaborative inference with modified Propose-Test-Release

**Abhishek Singh[1], Praneeth Vepakomma[1], Vivek Sharma[1,2,3], Ramesh Raskar[1]**
[1]MIT Media Lab, [2]MGH, Harvard Medical School, [3]Sony AI

## A Proofs

**Lemma A.1.** *For a given $f$ and for $d_{\mathcal{Y}} \leftarrow \ell_\alpha$ and $d_{\mathcal{X}} \leftarrow \ell_\beta$, $\Delta_{LS}(\mathbf{x}, R) \leq \mathcal{L}(f, \mathcal{N}(\mathbf{x}, R))$.*

*Proof.* Local sensitivity ($\Delta_{LS}$) for a sample $\mathbf{x}$ in a radius $R$ for a query $f$ is defined as:

$$\Delta_{LS}(\mathbf{x}, R) = \sup_{\mathbf{x}' \in \mathcal{N}(\mathbf{x}, R)} \frac{d_{\mathcal{Y}}(f(\mathbf{x}), f(\mathbf{x}'))}{d_{\mathcal{X}}(\mathbf{x}, \mathbf{x}')}$$

Local Lipschitz constant ($\mathcal{L}$) for a function $f$ and a neighborhood $\mathcal{N}$ is defined as:

$$\mathcal{L}^{\alpha, \beta}(f, \mathcal{N}) = \sup_{\mathbf{x}', \mathbf{x}'' \in \mathcal{N}} \frac{||f(\mathbf{x}') - f(\mathbf{x}'')||_\alpha}{||\mathbf{x}' - \mathbf{x}''||_\beta} \quad (\mathbf{x}' \neq \mathbf{x}'')$$

If $\mathcal{L}$ is defined around neighborhood $\mathcal{N}(\mathbf{x}, R)$ then the set over which local sensitivity is computed is a subset of the set over which local Lipschitz constant is estimated. Intuitively, local Lipschitz condition is for all possible pair of samples in the neighborhood while local sensitivity is for all samples with respect to the given sample. Since both conditions require a suprememum over the set, $\Delta_{LS}(\mathbf{x}, R) \leq \mathcal{L}(f, \mathcal{N}(\mathbf{x}, R))$. □

**Lemma A.2.** *Algorithm $\phi$ gives a lower bound on the query $\gamma$. That is, $\phi(\mathbf{x}, R) \leq \gamma(\mathbf{x}, R)$.*

*Proof.* The $\gamma$ query is defined as -

$$\gamma(\mathbf{x}, R) = \min_{\mathbf{x}' \in \mathcal{X}} \{ d_{\mathcal{X}}(\mathbf{x}, \mathbf{x}') \ \ s.t. \ \ \Delta_{LS}(\mathbf{x}', R) > \Delta_{LS}^p \}. \tag{1}$$

The $\phi$ query is defined as -

$$\phi(\mathbf{x}, R) = \frac{1}{2} \cdot \underset{R' \geq R}{\arg\max} \{ \mathcal{L}(f, \mathcal{N}(\mathbf{x}, R')) \leq \Delta_{LS}^p \} \tag{2}$$

For any given sample $\mathbf{x}$ and privacy parameters $(R, \Delta_{LS}^p)$ such that $s = \phi(\mathbf{x}, R)$, we know that $\forall \mathbf{x}' \in \mathcal{N}(\mathbf{x}, s)$

$$\mathcal{N}(\mathbf{x}', s) \subset \mathcal{N}(\mathbf{x}, 2s)$$
$$\implies \mathcal{L}(f, \mathcal{N}(\mathbf{x}', s)) \leq \mathcal{L}(f, \mathcal{N}(\mathbf{x}, 2s))$$

Based on eq 1, we know that $\mathcal{L}(f, \mathcal{N}(\mathbf{x}, 2s)) \leq \Delta_{LS}^p$ and hence $\forall \mathbf{x}' \in \mathcal{N}(\mathbf{x}, s)$,

$$\mathcal{L}(f, \mathcal{N}(\mathbf{x}', s)) \leq \Delta_{LS}^p$$

Therefore, $\Delta_{LS}(\mathbf{x}', R) \leq \Delta_{LS}^p$ and hence,

$$s \leq \gamma(\mathbf{x})$$

For the cases when there is not any feasible solution, $\phi$ returns 0 which is exactly the same answer for $\gamma$ query. This completes the proof. □

37th Conference on Neural Information Processing Systems (NeurIPS 2023).

**Lemma A.3.** *The query $\phi(\cdot)$ has a global sensitivity of $1$, i.e. $d_{abs}(\phi(\mathbf{x}, R), \phi(\mathbf{x}', R)) \leq d_{\mathcal{X}}(\mathbf{x}, \mathbf{x}')$*

*Proof.* We will prove the above argument through a contradiction. We will prove that for a fixed radius $R$ and any arbitrary point $\mathbf{x} \in \mathcal{X}$, the neighborhood spanned by $\phi(\mathbf{x}, R)$ can not be a proper superset for any neighborhood spanned by any other point $\phi(\mathbf{x}', R)$. More formally, we will prove, $\forall \mathbf{x}, \mathbf{x}' \in \mathcal{X}, \ \mathcal{N}(\mathbf{x}, \phi(\mathbf{x}, R)) \not\subset \mathcal{N}(\mathbf{x}', \phi(\mathbf{x}', R))$ and $\mathcal{N}(\mathbf{x}', \phi(\mathbf{x}', R)) \not\subset \mathcal{N}(\mathbf{x}, \phi(\mathbf{x}, R))$. Once proven, this argument allows us to specify the distance between $\mathbf{x}$ and $\mathbf{x}'$ with respect to $\phi(\mathbf{x}, R)$ and $\phi(\mathbf{x}', R)$.

Since the function $\phi(\mathbf{x}, R)$ returns the maximum possible value such that

$$\mathcal{L}(f, \mathcal{N}(\mathbf{x}, \phi(\mathbf{x}, R))) \leq \Delta_{LS}^{p}$$

Therefore, for any $\zeta > 0$

$$\mathcal{L}(f, \mathcal{N}(\mathbf{x}, \phi(\mathbf{x}, R) + \zeta)) > \Delta_{LS}^{p} \tag{3}$$

For contradiction, we assume that $\exists \mathbf{x}, \mathbf{x}' \in \mathcal{X}$ s.t. $\mathcal{N}(\mathbf{x}, \phi(\mathbf{x}, R)) \subset \mathcal{N}(\mathbf{x}', \phi(\mathbf{x}', R))$

$$\implies \exists \ \eta > 0 \ s.t. \ \mathcal{N}(\mathbf{x}, \phi(\mathbf{x}, R) + \eta) \subseteq \mathcal{N}(\mathbf{x}', \phi(\mathbf{x}', R)) \tag{4}$$

$$\implies \mathcal{L}(\mathcal{N}(\mathbf{x}, \phi(\mathbf{x}, R) + \eta)) \leq \Delta_{LS}^{p} \tag{5}$$

This leads to a contradiction between eq 3 and eq 5. Therefore, $\forall \mathbf{x}, \mathbf{x}' \in \mathcal{X}$,

$$\phi(\mathbf{x}, R) \leq \phi(\mathbf{x}', R) + d_{\mathcal{X}}(\mathbf{x}, \mathbf{x}')$$

Using symmetry argument, we can show that

$$d_{abs}(\phi(\mathbf{x}, R), \phi(\mathbf{x}', R)) \leq d_{\mathcal{X}}(\mathbf{x}, \mathbf{x}')$$

This completes the proof. $\qquad\square$

# B  Propose-Test-Release

DP mechanisms typically add noise based on the global sensitivity of a query. However, for several queries over various data distributions, average local sensitivity might be much lower than the global sensitivity. However, local sensitivity is data dependent hence the amount of noise introduced by a mechanism based on local sensitivity itself can reveal private information. Therefore, to add noise based on local sensitivity in a privacy preserving manner, [1] introduced PTR. Conceptually, the idea behind PTR is to propose an arbitrary upper bound on the true value of local sensitivity. This upper bound should be obtained privately otherwise the choice of upper bound itself can reveal private information. To test whether the proposed bound is correct, the mechanism performs a privacy-preserving testing of the upper bound. The test itself is a randomized algorithm due to privacy requirements. Therefore, it can have false positives and false negatives. If the test fails, the mechanism returns $\perp$ (no-answer). Otherwise, standard DP mechanism (ex. - Laplace) is applied to the query based on the proposed sensitivity (and not true local sensitivity). More formally, the algorithm proceeds in the following steps -

1. A proposal on upper bound of a query $q$ is fed as input for data $x$. Let us call it $\Delta_{LS}^{p}$.

2. The algorithm finds the closest point $x'$ to $x$ such that $\Delta_{LS}(x') > \Delta_{LS}^{p}$. Here $\Delta_{LS}$ refers to local sensitivity for the query $q$.

3. Let $\gamma = d_{\mathcal{H}}(x, x')$ and $\hat{\gamma} = \gamma + Lap(1/\epsilon)$

4. If $\hat{\gamma} \leq ln(1/\delta)/\epsilon$; return $\perp$

5. Else share data $x + Lap(\Delta_{LS}^{p}/\epsilon)$

**Computational Cost:** Depending on the query, this algorithm can incur significant computation cost. Especially in the Step 2, finding closest $x'$ can be impractical if the data space is high dimensional. Finally, for queries such as neural networks evaluating local sensitivity itself is not practical since it requires giving exact and correct solution to a non-convex optimization problem. Therefore, our framework relies on computing local lipschitz constant over a small neighborhood instead of local sensitvity over the complete data space.

**Algorithm 1:** Extended PTR algorithm for $(\epsilon, \delta, R)$-semantic neighborhood privacy

---

**Input:** $\mathbf{x} \in \mathcal{X}, \Delta_{LS}^{p} \in \mathbb{R}^{+}$;
**Params:** $\epsilon \in \mathbb{R}^{+}, \delta \in \mathbb{R}^{+}, R \in \mathbb{R}^{+}$;
**Init:** $\zeta \in \mathbb{R}^{+}$; *//For numerical stability, typically very small*
**Init:** $R_{min} = R, R_{max} \in \mathbb{R}^{+}$
**repeat**
   $R_{mid} = (R_{min} + R_{max})/2$;
   $r = \mathcal{L}(f, \mathcal{N}(\mathbf{x}, R))$; *//Compute local Lipschitz constant*
   **if** $r < \Delta_{LS}^{p}$ **then**
      $R_{min} = R_{mid}$;
   **else**
      $R_{max} = R_{mid}$
   **end if**
**until** $R_{max} > R_{min} + \zeta$
$\hat{r} = \frac{R_{min}}{2}$;
$\hat{R} \leftarrow \hat{r} + \mathsf{Lap}(1/\epsilon)$;
**if** $\hat{R} < ln(1/\delta)/\epsilon$ **then**
   **Return** $\perp$;
**else**
   **Return** $f(\mathbf{z}) + \mathsf{Lap}(\Delta_{LS}^{p}/\epsilon)$;
**end if**

---

## C Lipschitz Constant Estimation

We use the mixed integer programming based algorithm LipMip by [3] for computing the local Lipschitz constant. Their technique allows exactly computing the local Lipschitz constant of a neural network with ReLU non-linearities. Their key idea is to estimate the supremal norm of the jacobian of the neural network. Since ReLU networks do not allow for differentiability, LipMip uses clark jacobian to circumvent the issue and encode the optimization objective of obtaining local Lipschitz constant over a pre-defined neighborhood as a mixed integer programming problem. The neighborhood is specified as a hypercube with same dimension as points in the neighborhood. Their algorithm searches for feasible regions and minimizes the gap between lower and upper bound on the Lipschitz constant.

## D Experimental Details

Our experimental setup operates in three stages - i) Embedder training, ii) Obfuscator training, and iii) Private inference. Our codebase is available `https://drive.google.com/drive/folders/1DpHhS9u-Mpp3TVmTYiue7BKKUshyKw2w?usp=sharing` here for reproducability. We will release the code and all trained models publicly after the reviews. For all our experiments we use PyTorch([5]) with Nvidia-GeForce GTX TITAN GPU. We use $\beta$-VAE with $\beta = 5$ for the design of the *embedder*.

1. **Embedder Training:** We use embedding dimension as 8 for MNIST and FMNIST dataset. For the UTKFace dataset, we use embedding size as 10. We use Adam optimizer([4]) with a constant learning rate of 0.001. The VAE architecture for MNIST and FMNIST dataset is composed of three fully connected layers with non-linear activations and dropout.

2. **Obfuscator Training:** For ARL, we use $\alpha = 0.99$, for noisy regularization, we use $\sigma = 0.01$ and for contrastive regularization, we use $\lambda = 1.0$ with a *margin* of 25. All of these regularizations are trained jointly using Adam optimizer([4]).

3. **Private Inference:** In the this stage we use LipMip([3]) which is built upon Gurobi Optimizer([2]) for solving the Mixed-Integer programming formulation of local lipschitz constant estimation. For the metrics, we use $d_{\mathcal{X}}$ as infinity norm and $d_{\mathcal{Y}}$ as $\ell_1$-norm. For the privacy parameters, we use $\delta = 0.05$ and $R = 0.5$ for MNIST, $R = 0.2$ for FMNIST and

|  |  | $\epsilon = 1$ | $\epsilon = 2$ | $\epsilon = 5$ | $\epsilon = 7$ | $\epsilon = \infty$ |
|---|---|---|---|---|---|---|
| MNIST | LDP-Image | 0.1 | 0.1 | 0.1 | 0.3 | 0.99 |
|  | LDP-Embedding | 0.1075 | 0.1319 | 0.1927 | 0.3107 | 0.9096 |
|  | Adversarial | 0.514 | 0.751 | 0.891 | 0.912 | 0.9291 |
| FMNIST | LDP-Image | 0.1 | 0.1 | 0.1 | 0.1 | 0.92 |
|  | LDP-Embedding | 0.1012 | 0.1251 | 0.1708 | 0.2420 | 0.7798 |
|  | Adversarial | 0.371 | 0.554 | 0.678 | 0.722 | 0.781 |
| UTKFace | LDP-Image | 0.52 | 0.52 | 0.52 | 0.52 | 0.89 |
|  | LDP-Embedding | 0.5040 | 0.535 | 0.5757 | 0.6375 | 0.7246 |
|  | Adversarial | 0.65 | 0.69 | 0.718 | 0.724 | 0.73 |

Table 1: Comparison between LDP and Adversarial Representation Learning: Using our proposed framework we compare the utility of LDP and ARL across different values of the privacy parameter $\epsilon$.

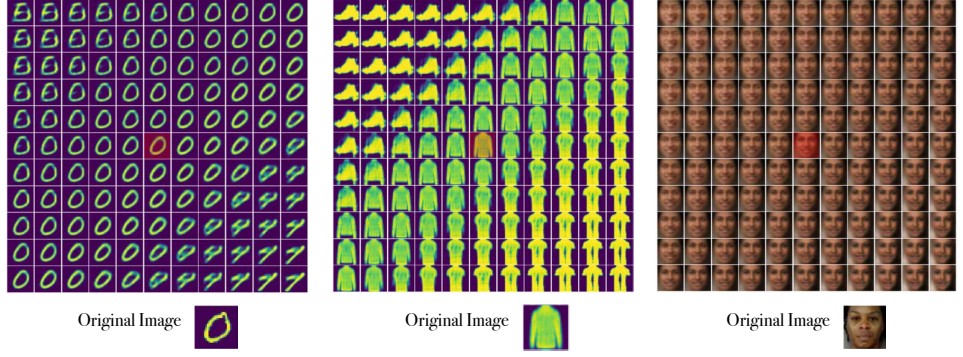

| Original Image | Original Image | Original Image |
|---|---|---|

Figure 1: Neighborhood for different image datasets. The center image (in red) is the reconstruction of the original image with nearby images sampled from the embedding. Note that there are multiple dimensions and we have illustrated interpolation for only one here.

$R = 0.1$ for UTKFace. The choice of different $R$ was based on visualizing samples from the training set and evaluating how far similar looking samples lie in the embedding space.

| SSIM | MNIST |  |  |  |  |  | UTKFace |  |  |  |  |  |
|---|---|---|---|---|---|---|---|---|---|---|---|---|
|  | 0.1 | 0.2 | 0.4 | 0.6 | 0.8 | 1.0 | 0.1 | 0.2 | 0.4 | 0.6 | 0.8 | 1.0 |
| $\epsilon=1$ | 0.558 | 0.427 | 0.287 | 0.231 | 0.201 | 0.179 | 0.481 | 0.443 | 0.426 | 0.425 | 0.422 | 0.421 |
| $\epsilon=2$ | 0.648 | 0.553 | 0.414 | 0.328 | 0.275 | 0.229 | 0.507 | 0.475 | 0.442 | 0.439 | 0.428 | 0.426 |
| $\epsilon=5$ | 0.702 | 0.655 | 0.580 | 0.499 | 0.434 | 0.380 | 0.519 | 0.5037 | 0.481 | 0.465 | 0.451 | 0.447 |
| $\epsilon=10$ | 0.710 | 0.700 | 0.656 | 0.610 | 0.560 | 0.517 | 0.519 | 0.5169 | 0.507 | 0.494 | 0.485 | 0.476 |

Table 2: SSIM metric for reconstruction attack with varying $R$ and $\epsilon$

| $\ell_1$ | MNIST | | | | | | UTKFace | | | | | |
|---|---|---|---|---|---|---|---|---|---|---|---|---|
| | 0.1 | 0.2 | 0.4 | 0.6 | 0.8 | 1.0 | 0.1 | 0.2 | 0.4 | 0.6 | 0.8 | 1.0 |
| $\epsilon$=1 | 0.0775 | 0.0968 | 0.1145 | 0.1204 | 0.1232 | 0.1249 | 0.1304 | 0.1532 | 0.1706 | 0.1731 | 0.1776 | 0.1782 |
| $\epsilon$=2 | 0.0642 | 0.078 | 0.0983 | 0.1091 | 0.1153 | 0.1196 | 0.1126 | 0.1312 | 0.1543 | 0.1645 | 0.1699 | 0.1732 |
| $\epsilon$=5 | 0.0565 | 0.0634 | 0.0749 | 0.0859 | 0.0959 | 0.1031 | 0.1017 | 0.1089 | 0.125 | 0.1388 | 0.148 | 0.1546 |
| $\epsilon$=10 | 0.0549 | 0.0569 | 0.0631 | 0.0697 | 0.0771 | 0.0837 | 0.1001 | 0.1027 | 0.1091 | 0.1182 | 0.1255 | 0.1321 |

Table 3: $\ell_1$ metric for reconstruction attack with varying $R$ and $\epsilon$

| $\ell_2$ | MNIST | | | | | | UTKFace | | | | | |
|---|---|---|---|---|---|---|---|---|---|---|---|---|
| | 0.1 | 0.2 | 0.4 | 0.6 | 0.8 | 1.0 | 0.1 | 0.2 | 0.4 | 0.6 | 0.8 | 1.0 |
| $\epsilon$=1 | 0.0471 | 0.0613 | 0.0762 | 0.0817 | 0.0847 | 0.0867 | 0.029 | 0.0385 | 0.0464 | 0.0476 | 0.0492 | 0.0498 |
| $\epsilon$=2 | 0.0363 | 0.0478 | 0.0631 | 0.0716 | 0.0773 | 0.0814 | 0.0223 | 0.029 | 0.0387 | 0.0433 | 0.0454 | 0.0471 |
| $\epsilon$=5 | 0.03 | 0.0354 | 0.0449 | 0.0537 | 0.0618 | 0.0672 | 0.0189 | 0.021 | 0.027 | 0.032 | 0.0358 | 0.0388 |
| $\epsilon$=10 | 0.0291 | 0.0305 | 0.0352 | 0.0407 | 0.0467 | 0.0518 | 0.0184 | 0.0192 | 0.0212 | 0.0243 | 0.0271 | 0.0294 |

Table 4: $\ell_2$ metric for reconstruction attack with varying $R$ and $\epsilon$

| PSNR | MNIST | | | | | | UTKFace | | | | | |
|---|---|---|---|---|---|---|---|---|---|---|---|---|
| | 0.1 | 0.2 | 0.4 | 0.6 | 0.8 | 1.0 | 0.1 | 0.2 | 0.4 | 0.6 | 0.8 | 1.0 |
| $\epsilon$=1 | 61.87 | 60.70 | 59.74 | 59.45 | 59.29 | 59.19 | 64.42 | 63.18 | 62.36 | 62.25 | 62.10 | 62.05 |
| $\epsilon$=2 | 63.04 | 61.80 | 60.57 | 60.02 | 59.68 | 59.47 | 65.59 | 64.43 | 63.16 | 62.66 | 62.45 | 62.29 |
| $\epsilon$=5 | 63.88 | 63.15 | 62.09 | 61.28 | 60.67 | 60.29 | 66.30 | 65.84 | 64.75 | 63.99 | 63.50 | 63.15 |
| $\epsilon$=10 | 64.06 | 63.82 | 63.19 | 62.54 | 61.91 | 61.44 | 66.43 | 66.25 | 65.80 | 65.21 | 64.72 | 64.36 |

Table 5: PSNR metric for reconstruction attack with varying $R$ and $\epsilon$