# OpenReview forum: "Posthoc privacy guarantees for collaborative inference with modified Propose-Test-Release"
_NeurIPS.cc/2023/Conference — NeurIPS 2023 poster_

### Official Review · Reviewer_UmFA · 2023-07-05

**Soundness:** 2 fair
**Presentation:** 3 good
**Contribution:** 2 fair
**Rating:** 3
**Confidence:** 5

**Summary:**

The paper proposed to protect data privacy in collaborative inference settings.  Specifically, the paper presents a formal way to capture the obfuscation of an image using adversarial representation learning, a well-studied technique. The paper does so by using a metric-based differential privacy notion. Since the metric is difficult to compute, the paper proposes a way to estimate Lipschitz constant using an estimation technique from DP.

**Strengths:**

+ The studied problem is important

+ The paper is easy to follow

+ Formal privacy guarantees

**Weaknesses:**

- Threat model is unrealistic
- Lack of evaluation against advanced attacks
- More clarifications are needed
- Experimental result is not convincing


**Questions:**

My big concern is that the threat model requires knowing the classifier (by the obfuscator/defender) which makes it unable to protect all attacks and impracticable in many cases. For instance, the data owner has a stringent requirement about his data privacy, e.g., not
 letting the defender knows the learning task.  In other words, when knowing the classifier, why would the data owner send his data for classification to the server in the first place?

“most of these works analyze specific obfuscation techniques and lack formal privacy definitions” This claim seems inaccurate.
For instance,  Zhao et al. 2020a indeed showed formal privacy guaranteed information leakage. What are the key differences? How do you compare your theoretical results with theirs?

Table 1 shows that ARL-C-N is even worse than Encoder in several cases. Why does this happen? How can you claim that ARL-C-N performs the best among all baselines?

Two recent works [a,b] propose advanced attacks against ``representation learning” based defense. I would like to see the performance of the proposed ARL-C-N .

[a] Song et al., Overlearning reveals sensitive attributes. In ICLR, 2020
[b] Balunovic et al., Bayesian framework for gradient leakage. ICLR’22

Balle et al. [c] also showed theoretical results on data reconstruction via DP technique. What are the key technical differences?

[c] Balle et al., “Reconstructing training data with informed adversaries,” in IEEE SP, 2022.



**Limitations:**

Yes.

---

> ### Author Rebuttal · Authors · 2023-08-09
>
> We thank the reviewer for the feedback. We respectfully disagree with several points mentioned by the reviewer and have tried our best to address those by quoting them inline.
> First, we would like to clarify “collaborative inference” as studied in several existing papers[1-5] including ours.
> In collaborative inference, an embedding from an intermediate layer is shared by a client to a server and several existing works have proposed (informally private) obfuscation techniques to prevent these embeddings from input reconstruction attacks. Our work presents a framework to give a formal privacy guarantee for these obfuscation techniques.
>
> **My big concern is that the threat model requires knowing the classifier (by the obfuscator/defender)**
>
> The threat model does not require the defender to know the classifier. In fact, threat models (including ours) do not impose any restriction on defenders, they only focus on the capability of the adversary.
>
> **why would the data owner send his data for classification to the server in the first place?**
>
> There isn’t any strict requirement for the data owner to know the classifier. It is the training of the obfuscator that requires access to the classifier. However, such training can be performed by the service provider or a third party. We would like to emphasize that our work only focuses on inference and not training.
>
> A typical use-case for collaborative inference is that a cloud-based service provider company has a model which they want their users to query with their private data. However, the user can not run it on their own device (either due to the model size or the proprietary nature of the model). Therefore, the cloud-based service provider gives the end user an obfuscator so that it can share its obfuscated data with the service provider.
>
> **How do you compare your theoretical results with Zhao et al.2020a?**
>
> Zhao et al. 2020a study attribute obfuscation. In contrast, we study reconstruction attacks over the input data. We highlighted this important distinction in line 73 by stating -
> > A majority of the CI techniques protect either a sensitive attribute or reconstruction of the input. We only consider sensitive input in this work.
>
> In contrast, quoting Zhao et al. on Page 3, Lines 3 and 4 -
>
> > To simplify the exposition, we mainly discuss the setting where $X \subseteq R^d$, $Y = A = \{0, 1\}$, but the underlying theory and methodology could easily be extended to the categorical case as well.
>
> Therefore, they study binary (and categorical) sensitive attributes. Hence, Attribute Inference Advantage (their privacy definition) can not be extended to reconstruction privacy in the input space of data.
>
> **Table 1 shows that ARL-C-N is even worse than Encoder in several cases. Why does this happen? How can you claim that ARL-C-N performs the best among all baselines?**
>
> ARL-C-N is generally better than other techniques (best performing for 8 out of 12 combinations of datasets and privacy budgets). However, we agree this statement should be made more precisely. The “Informal” column is just for reference and shouldn’t be considered formally private. Considering that, the encoder is only higher than ARL-C-N for $\epsilon=10$ (except $\epsilon=5$ in UTKFace). This is due to the high local sensitivity of the Encoder (bad for privacy-utility trade-off) that gets discounted when the epsilon value is high.
>
> **Two recent works [a,b] propose advanced attacks against ``representation learning” based defense. I would like to see the performance of the proposed ARL-C-N.**
>
> We would like to clarify that ARL-C-N is not our proposed technique and is simply yet another defense mechanism. We have evaluated three popular classes of collaborative inference techniques – ARL[2,3,4], C[5], and N[6]. ARL-C-N is an ablation where all three regularizations are active. We have ablated all 2^3 combinations in the experiments and ARL-C-N is one of them. Song et al. attack focuses on inferring sensitive attributes while our attack is about reconstructing the input data from its embedding during inference. However, both attacks use supervised learning to train an attacker, and hence the training algorithm for the attacker is the same as the one we employed in Table 2 of the supplementary.
>
> While Balunovic et al. attack representation learning techniques, their attacks rely on access to gradients because they focus on Federated Learning. In contrast, gradients are not involved at all in our context because we focus on Collaborative Inference.
>
> **Balle et al. [c] also showed theoretical results on data reconstruction via the DP technique.**
>
> Balle et al. are also out of scope from our line of work. Their focus is on reconstructing training data by accessing the parameters of a trained model. Quoting the first line of their abstract -
> > Given access to a machine learning model, can an adversary reconstruct the model’s training data?
>
> In contrast, we study the problem of reconstruction attacks over input data during the inference (not training) stage.
>
> References
> 1. Yang, Mengda, et al. "Measuring Data Reconstruction Defenses in Collaborative Inference Systems." Advances in Neural Information Processing Systems 35 (2022): 12855-12867.
> 2. Singh, Abhishek, et al. "Disco: Dynamic and invariant sensitive channel obfuscation for deep neural networks." Proceedings of the IEEE/CVF Conference on Computer Vision and Pattern Recognition. 2021.
> 3. Xiao, Taihong, et al. "Adversarial learning of privacy-preserving and task-oriented representations." Proceedings of the AAAI Conference on Artificial Intelligence. Vol. 34. No. 07. 2020.
> 4. Osia, Seyed Ali, et al. "A hybrid deep learning architecture for privacy-preserving mobile analytics." IEEE Internet of Things Journal 7.5 (2020): 4505-4518.
> 5. Mireshghallah, Fatemehsadat, et al. "Not all features are equal: Discovering essential features for preserving prediction privacy." Proceedings of the Web Conference 2021. 2021.

---

> > ### Comment · Reviewer_UmFA · 2023-08-15
> > **Response to Authors' Rebuttal**
> >
> > Thanks for the response. However, several of my main comments are not fully addressed.
> >
> >
> > **Threat model: We would like to emphasize that our work only focuses on inference and not training**
> >
> > The goal of the considered data reconstruction defense should be that the encoding cannot be used to infer the private data, while the encoding itself should maintain high utility.  The authors said they focus on inference and not training, but the defender needs to perform obfuscator training. I am confused what the authors are exactly trying to protect?
> >
> >
> > **Balle et al. are also out of scope from our line of work…. In contrast, we study the problem of reconstruction attacks over input data during the inference (not training) stage.**
> >
> > Balle et al. 2022 focus on reconstructing training data from informal adversaries who know all the training data except one. The adversaries are also assumed to know the released model and thus know all the intermediate representations.
> > I do not know why the authors cannot compare their theoretical results with Balle et al.’s?  Note that Theorem 4.4 and Theorem 2 in Balle et al. also look similar.

---

> > > ### Author Response · Authors · 2023-08-15
> > > **Response to reviewer's comment**
> > >
> > > We thank the reviewer for responding. We will keep the response short as we now understand the point of confusion.
> > >
> > > We are trying to protect the leakage of inference (test) samples that are only used during the prediction stage. Our work is agnostic to how the obfuscator gets trained because training data is not private.
> > >
> > > As a concrete example - consider a single image $x$, obfuscator computes $z=f(\theta,x)$ where $f(\theta,\cdot)$ refers to a neural network trained on dataset $D$. This $z$ is shared with an untrusted party. Our framework only protects the reconstruction of $x$ from $z$. We do not study the leakage of $D$ from $z$.
> > >
> > > Please let us know if it doesn't address the concerns and we can expand further on the comments.

---

> > > > ### Comment · Reviewer_UmFA · 2023-08-21
> > > > **Response to Authors**
> > > >
> > > > Thanks for your reply. However, your did not address my second comment. That is, I am still not convinced why Balle et al. 2022 cannot be used to test your framework. A very similar framework is also proposed in Guo et al.
> > > >
> > > > Guo et al. Bounding Training Data Reconstruction in Private (Deep) Learning. ICML 2022

---

> > > > > ### Author Response · Authors · 2023-08-21
> > > > > **Response to the reviewer**
> > > > >
> > > > > We thank the reviewer for responding.
> > > > > - Balle et al. 2022 can not be used directly because their attacker reconstructs training data by querying a model trained on private data.
> > > > >   - Our work does not consider training data to be private and hence the model does not leak private information.
> > > > >   - The closest we can get to their attack is by training an attacker model to reconstruct data from embedding. We have tested against such an attacker in the supplementary Table 2.
> > > > > - Thanks for pointing to Guo et al. We briefly discussed it in the paper and will add a more detailed comparison with their work in the final manuscript. In summary, once again, they focus on the reconstruction of training data of an ML model by querying it. We do not privatize training data or models.
> > > > >   - Specifically - the first line of their threat model states: _We focus on formalizing data reconstruction attacks and deriving semantic guarantees against data reconstruction attacks for private learning algorithms._
> > > > >   - In contrast, our work is not focused on private learning but on private inference.
> > > > > - The common theme of both of these works in relation to ours is that they show reconstruction attacks can be prevented in the cases where membership inference attack is still possible. However, the threat models are different for our and their work. For their mechanisms, multiple private samples are used for training and it is the model that leaks private information. In our mechanism, only a single private sample is used, and its embedding leaks private information about test data during inference that we protect.

---

### Official Review · Reviewer_ybRi · 2023-07-05

**Soundness:** 3 good
**Presentation:** 3 good
**Contribution:** 2 fair
**Rating:** 7
**Confidence:** 2

**Summary:**

A method is proposed to encode inputs to a server that performs inference, to limit the ability of the service provider to infer detailed information about the input. A formal privacy guarantee, inspired by differential privacy, quantifies the amount of leakage. This metric is related to the local Lipschitz constant, which relation is used in the test of a propose-test-release mechanism to ensure local privacy.

**Strengths:**

The concepts introduced are original as far as I know and the exposition is mostly clear.

**Weaknesses:**

The primary weakness in my opinion is in the practical utility of the proposed method. If I understand correctly, the method does not provide local differential privacy (indeed the authors argue this is impossible for a system designed to return useful classification or functions of the input) but merely prevents the inference of detailed information. Specifically, it attempts to obfuscate any information that is not necessary for the task. It seems to me much of the time this would not guarantee "privacy" in the sense that most people would want. Suppose for example that we are working with linguistic input. Maybe it is a question answering service, or machine translation. Users of a purportedly private system would expect that the service would not have access to (not be able to examine or log) the specific question being answered, or the text being translated. Clearly that is incompatible with providing sufficient utility-- something like holomorphic encryption or trusted execution environments would be needed for that. So I think the paper would be stronger if it were motivated with some concrete applications where a user would reasonably be satisfied with the specific type of privacy afforded by this approach.

**Questions:**

What are some concrete applications where a user would reasonably be satisfied with the specific type of privacy afforded by this approach?

**Limitations:**

Limitations are discussed. No potential negative societal impact.

---

> ### Author Rebuttal · Authors · 2023-08-09
>
> We thank the reviewer for the feedback. We agree that tasks which require granular information from data such as translation would not be possible with collaborative inference approaches. However, plenty of use cases exist where all the information is not necessarily required to produce an answer. These uses-cases typically produce a significantly smaller dimensional output for high dimensional input. Examples include -
>
> - Image, Graphs, or Text classification tasks - A significant amount of information can be thrown away while still being able to compute the answer.
> - Vector databases for language model - With the improved performance of language models, several use cases for vector databases have arisen such as document retrieval, product recommendation, etc. For example - Authors of [1] tried to apply privacy to classification tasks by using traditional local DP, and hence, they had to resort to extremely large epsilons (around 200) to achieve a reasonable utility. We believe such a technique could benefit from the privacy definition proposed in our work.
>
> **References** -
> 1. Timour Igamberdiev and Ivan Habernal. 2023. DP-BART for Privatized Text Rewriting under Local Differential Privacy. In Findings of the Association for Computational Linguistics: ACL 2023, pages 13914–13934, Toronto, Canada. Association for Computational Linguistics.

---

> > ### Comment · Reviewer_ybRi · 2023-08-14
> > **Acknowledgement of rebuttal**
> >
> > Thanks for answering my question. I will maintain my rating.

---

### Official Review · Reviewer_MzZs · 2023-07-10

**Soundness:** 3 good
**Presentation:** 3 good
**Contribution:** 3 good
**Rating:** 6
**Confidence:** 4

**Summary:**

This paper proposes a method for evaluating the privacy guarantees provided through adversarial representation learning. More precisely, the framework proposed is built on the Propose-Test-Release paradigm (PTR) as well as the d_x-privacy metric. The primary objective is to be able to characterize the privacy risks in terms of reconstruction against the adversarial representation at inference time.

**Strengths:**

The paper is well-written and the authors have a done job at explaining the limits of previous works on assessing the security of adversarial representation learning. However, the definition of the Lipschitz constant should also be part of the main body of the paper rather than being in the supplementary material.

The move from differential privacy to d_x-privacy enables to make the estimation of the local sensitivity tractable, which circumvents some of the limits of the PTR framework.

The proposed approach is tested on three diverse datasets and the results obtained demonstrate that it is comparable in performance to other methods from the state-of-the-art.

**Weaknesses:**

There are several possible way to implement a LDP mechanism. I suggest to the authors to mention which one they have implemented (for instance is it randomized response) and also to try a different one to see if the results they obtained are dependent or not of the particular mechanisms evaluated.

Even if the focus of the current approach is to prevent reconstruction attack, I suggest to also perform additional experiments to measure the success of membership inference attacks. This would help to fully characterize the privacy-utility trade-off of the proposed method.

A few typos :
-« hamming distance » -> « Hamming distance »
-« laplace mechanism » -> « Laplace mechanism »
-The following sentence should be completed : « as we show in Sec ?? »

**Questions:**

-It would be helpful to understand the limits of differential privacy if the authors could discuss in a bit more details the impossibility results of instance encoding (reference 7).

-Please see also the weaknesses section.

**Limitations:**

Ok.

---

> ### Author Rebuttal · Authors · 2023-08-09
>
> We thank the reviewer for the feedback and address the concerns as follows -
>
> **LDP mechanisms and limits of differential privacy** - We note that our LDP experiment is only for the purpose of illustrating the gap in utility. The protection by LDP is so high that it is unreasonable to expect any utility (as we discuss below) in the context of Collaborative Inference (CI). Having said that, our LDP implementation is based on the Laplace mechanism with clipping. In order to bound the sensitivity of a sample we clip each dimension within a fixed range and add Laplace noise to the sample based on its sensitivity (clipping range and dimensions).
> A majority of the LDP mechanisms (such as randomized response, CountSketch, etc.) are designed for aggregation queries like count, mean, etc. However, in CI we only have a single sample over which post-processing (classifier) is applied and the answer is returned to the user.
>
> This also leads to the impossibility of applying DP directly in this context as DP would require the obfuscated embedding from any two samples to be indistinguishable (up to $e^\epsilon$) while the task of classification requires samples belonging to different classes to be distinguishable. A stronger version of this claim is given in instance encoding that says a dataset of such obfuscated embeddings can either have high utility or high privacy but not both. We address this impossibility by changing the metric space from hamming (as traditionally done in DP) to a general metric and using an l-norm ball around the input such that any two samples within that l-norm ball are indistinguishable. For example - Igamberdiev and Habernal apply LDP for Privatized Text Rewriting – a classification task, and hence, they had to use extremely large epsilons (around 200) to achieve a reasonable utility. We will make sure to include a more detailed discussion in the extra page of the camera-ready version.
>
> **Comparison with membership attacks** - We believe a membership attack is not possible because aggregate statistics are not shared. Specifically, a membership inference attack would require access to an aggregate statistic (such as mean or a ML model) and the attacker would have to guess whether a given sample was used to compute that statistic. In contrast, a low-dimension projection of a private input is shared. In that view, there isn’t any “model” to query for membership inference.
>
> **References** -
> Timour Igamberdiev and Ivan Habernal. 2023. DP-BART for Privatized Text Rewriting under Local Differential Privacy. In Findings of the Association for Computational Linguistics: ACL 2023, pages 13914–13934, Toronto, Canada. Association for Computational Linguistics.

---

> > ### Comment · Reviewer_MzZs · 2023-08-19
> > **Thanks for your response**
> >
> > With respect to LDP, while it is true that usually it is used to randomize profiles of individuals and then compute aggregate information it can still be view as an obfuscated version of the original profile and could be used also as as input to receive a personalized prediction. It would be great if the authors could look at existing LDP mechanisms to see if some of them could possibly acts as an obfuscation in a similar manner as collaborative inference.
> >
> > For membership inference attacks, as the obfuscator is learnt through a training process, I still think it makes sense to evaluate whether such an attack is possible by querying the obfuscator. In particular, it is possible that there is a difference of behaviour between member and non-member with respect to the obsfucation that makes it possible to conduct a membership inference attack.

---

> > > ### Author Response · Authors · 2023-08-21
> > > **Thanks for your comments**
> > >
> > > We thank the reviewer for the response.
> > > - We agree with the reviewer that other LDP mechanisms can be also applied similarly to how we have baselined with the truncated Laplace mechanism.
> > >   - Following the taxonomy discussed in Wang et al, we believe perturbations performed for frequency and mean estimation (typically by applying binary/unary encoding, sketching, or hashing of data) are not directly applicable.
> > >   - However, projection-based mechanisms typically used for estimating k-way marginals of high-dimensional data can be applied. We will include these perturbation mechanisms in addition to the truncated Laplace mechanism for our baselines in Supplementary Table 1.
> > > - While the obfuscator is learned through a training process, that training data is not private. Nevertheless, an interesting analysis would be to observe if these obfuscator models trained for privatizing inference data can also demonstrate different behavior for member and non-member training data points. Since it is orthogonal to the focus of our work and limited time, we could not do such an analysis during the rebuttal phase but we will include it in the final manuscript.
> > >
> > > References
> > > 1. Wang, T., Zhang, X., Feng, J., & Yang, X. (2020). A comprehensive survey on local differential privacy toward data statistics and analysis. Sensors, 20(24), 7030.

---

### Official Review · Reviewer_uj9h · 2023-07-17

**Soundness:** 3 good
**Presentation:** 3 good
**Contribution:** 3 good
**Rating:** 6
**Confidence:** 4

**Summary:**

This paper is concerned with the development of privacy-preserving collaborative inference. A collaborative inference framework allows users to create a privacy-preserving encoding of their data, which in turn can be used in place of real private inputs when interacting with machine learning-based services. The promise of privacy-preserving collaborative inference is that an adversary observing an encoding should not be able to recover the original input or a sensitive attribute of the original input. The authors propose a new framework to formally evaluate the privacy guarantees of such an encoding against reconstruction attacks that seek to recover the original input.

**Strengths:**


- The paper made several technical improvements to render the proposed solution useful in practice
- The proposed solution has been evaluated extensively

**Weaknesses:**


- Lack of discussion on the following items:
	- applicability of the framework to tabular datasets
	- guarantees of the proposed approach if the adversary has white box access to the obfuscator
- The paper focuses on collaborative inference as a solution for input privacy at inference time. The related work on secure inference could benefit from discussions on other approaches such as secure multi-party computation and homomorphic encryption.

**Questions:**

Please provide more discussion on the following elements
	- applicability to tabular data
	- the robustness of the framework to adversaries that have white-box access to the obfuscator
	- comparison of CI-based approaches to other secure inference techniques relying on secure multiparty computation and homomorphic encryption

---

> ### Author Rebuttal · Authors · 2023-08-09
>
> We thank the reviewer for providing the feedback. We address the three main discussion points brought up by the reviewer below -
>
> **Application to tabular data** - Our framework is agnostic to the data modality. From the point of view of privacy, the input needs to be a vector so that it can be privatized by an obfuscator. From a utility point of view, the obfuscation model should have low sensitivity.
>
> **The robustness of the framework to adversaries with white-box access to the obfuscator** - Our threat model assumes the adversary has white-box access to the obfuscator as discussed in Sec 3.1 and hence the privacy guarantees hold equally well.
>
> **Comparison between CI and secure inference techniques** - Thanks for pointing this out. We will update our Related Work section with a more detailed discussion and references to secure inference techniques.
> Secure inference[1,2,3,4] provides a high level of privacy (that is every bit in the input data is almost completely indistinguishable to an attacker from a randomly generated bit). However, as the computation performed by the service provider gets more sophisticated, it comes at the expense of higher computation and communication costs.
>
> In contrast, CI is computationally cheaper and does not incur any extra communication costs. However, CI offers less privacy as the input sample is only indistinguishable in a neighborhood around the input.
> CI is also more flexible because secure inference techniques need infrastructure implemented by the service provider, but obfuscation methods are user-controlled. Furthermore, shared embeddings in CI are in the plaintext space and hence amenable to additional analysis such as downstream training as performed in Split Learning and Vertical Federated Learning.
>
> **References** -
> 1. Wagh, Sameer, Divya Gupta, and Nishanth Chandran. "Securenn: Efficient and private neural network training." Cryptology ePrint Archive (2018).
> 2. Lou, Qian, et al. "Falcon: Fast spectral inference on encrypted data." Advances in Neural Information Processing Systems 33 (2020): 2364-2374.
> 3. Reagen, B., Choi, W.S., Ko, Y., Lee, V.T., Lee, H.H.S., Wei, G.Y. and Brooks, D., 2021, February. Cheetah: Optimizing and accelerating homomorphic encryption for private inference. In 2021 IEEE International Symposium on High-Performance Computer Architecture (HPCA) (pp. 26-39). IEEE.
> 4. Lee et al., 2022, June. Low-Complexity Deep Convolutional Neural Networks on Fully Homomorphic Encryption Using Multiplexed Parallel Convolutions. In International Conference on Machine Learning (pp. 12403-12422). PMLR.

---

> > ### Comment · Reviewer_uj9h · 2023-08-22
> >
> > Thank you for your clarifications. I will keep my original score.

---

### Decision · Program_Chairs · 2023-09-21

**Decision:**

Accept (poster)

**Comment:**

Most reviewers agreed that the paper should be accepted, although there were some common points of confusion that the authors should clarify in the final version. In particular, they should emphasize that only the privacy of data provided during inference, and not the privacy of training data, is protected by their mechanism.